# MahiNet: A Neural Network for Many-Class Few-Shot Learning with Class Hierarchy

## Abstract

We study many-class few-shot (MCFS) problem in both supervised learning and meta-learning scenarios. Compared to the well-studied many-class many-shot and few-class few-shot problems, MCFS problem commonly occurs in practical applications but is rarely studied. MCFS brings new challenges because it needs to distinguish between many classes, but only a few samples per class are available for training. In this paper, we propose "memory-augmented hierarchical-classification network (MahiNet)" for MCFS learning. It addresses the "many-class" problem by exploring the class hierarchy, e.g., the coarse-class label that covers a subset of fine classes, which helps to narrow down the candidates for the fine class and is cheaper to obtain. MahiNet uses a convolutional neural network (CNN) to extract features, and integrates a memory-augmented attention module with a multi-layer perceptron (MLP) to produce the probabilities over coarse and fine classes. While the MLP extends the linear classifier, the attention module extends a KNN classifier, both together targeting the "few-shot" problem. We design different training strategies of MahiNet for supervised learning and meta-learning. Moreover, we propose two novel benchmark datasets "*mcfs*ImageNet" (as a subset of ImageNet) and "*mcfs*Omniglot" (re-splitted Omniglot) specifically for MCFS problem. In experiments, we show that MahiNet outperforms several state-of-the-art models on MCFS classification tasks in both supervised learning and meta-learning scenarios.

## 1 Introduction

The representation power of deep neural networks (DNN) has dramatically improved in recent years, as deeper, wider and more complicated DNN architectures (He et al., 2016; Huang et al., 2017) have emerged to match the increasing computation power of new hardwares. Although this brings hope for complex tasks that could be hardly solved by previous shallow models, more training data is usually required. Hence, the scarcity of annotated data has become a new bottleneck for training more powerful DNNs. For example, in image classification, the number of candidate classes can easily range from hundreds to tens of thousands (i.e., many-class), but the training samples available for each class can be less than $100$ (i.e., few-shot). Additionally, in life-long learning, models are always updated once new training data becomes available, and those models are expected to quickly adapt to new classes with a few training samples. This "many-class few-shot" problem is very common in various applications, such as image search, robot navigation and video surveillance.

Although enormous previous works have shown the remarkable power of DNN when "many-class many-shot" training data is available, their performance degrades dramatically when each class only has a few samples available for training. In practical applications, acquiring samples of rare species is usually difficult and often expensive. In these few-shot scenarios, the model's capacity cannot be fully utilized, and it becomes much harder to generalize the model to unseen data. Recently, several approaches have been proposed to address the few-shot learning problem. Most of them are based on the idea of "meta-learning", which trains a meta-learner that can generalize to different tasks. For classification, each task targets a different set of classes. Meta-learning can be categorized into two types: methods based on "learning to optimize", and methods based on metric learning. The former type adaptively modifies the optimizer (or some parts of it) applied to the training process. It includes methods that incorporate an RNN meta-learner (Andrychowicz et al., 2016; Li & Malik, 2017; Ravi & Larochelle, 2017), and model-agnostic meta-learning (MAML) methods aiming to learn a

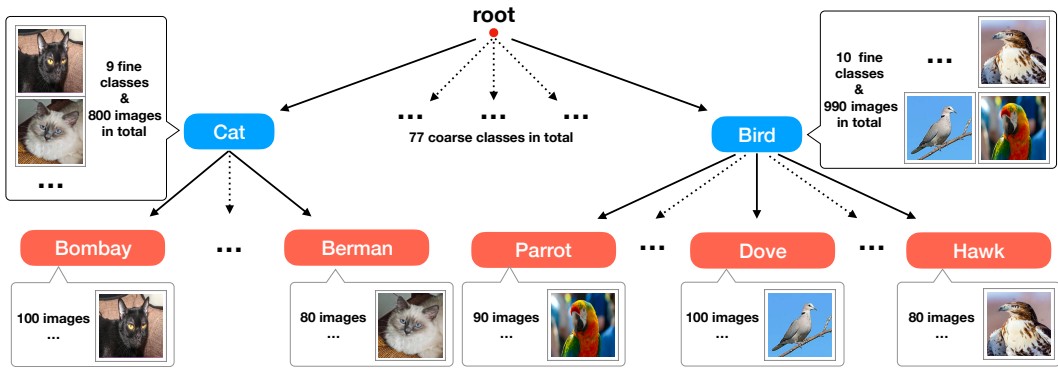

Figure 1: The MCFS problem with class hierarchy information. There are a few coarse classes (blue), but each coarse class contains a large number of fine classes (red), and the total number of fine classes is large. Only a few training samples are available for each fine class. The goal is to train a classifier to generate a prediction over all fine classes. In meta-learning, each task is an MCFS problem sampled from a certain distribution. The meta-learner's goal is to help train a classifier for any sampled task with better adaptation to few-shot data.

generally compelling initialization (Finn et al., 2017). The latter type learns a similarity/distance metric (Vinyals et al., 2016) or a support set of samples (Snell et al., 2017) that can be generally used to build KNN classifiers for different tasks.

Instead of using meta-learning, some other approaches, such as Douze et al. (2018), address the few-shot learning problem through data augmentation by generating artificial samples for each class. However, most existing few-shot learning approaches only focus on "few-class" case (e.g., 5 or 10) per task, and performance usually collapses when the number of classes grows to hundreds or thousands. This is because the samples per class no longer provide enough information to distinguish them from other possible samples within a large number of other classes. And, in real-world problems, tasks are usually complicated involving many classes.

Table 1: Targeted problems of different methods.

|  | many-class | few-class |
|---|---|---|
| few-shot | **MahiNet (ours)** | MAML, Matching Net, *etc.* |
| many-shot | ResNet, DenseNet, Inception, VGG, *etc.* | |

Fortunately, in practice, class hierarchy is usually available or cheaper to obtain. As shown in Figure 1, coarse class labels might reveal the relationships among the targeted fine classes. Moreover, the samples per coarse class are sufficient to train a reliable coarse classifier, whose predictions are able to narrow down the candidates for fine classes. For example, a sheepdog with long hair could be easily mis-classified as mop when training samples of sheepdog are insufficient. However, if we could train a reliable dog classifier, it would be much simpler to predict an image as a sheepdog than a mop given a correct prediction of the coarse class as "dog". Hence, class hierarchy might provide weakly supervised information to help solve the "many-class few-shot (MCFS)" problem.

## 1.1 OUR APPROACH

In this paper, we study how to explore the class hierarchy to solve MCFS problem in both traditional supervised learning and in meta-learning. We develop a DNN architecture "memory-augmented hierarchical-classification networks (MahiNet)" that can be applied to both learning scenarios. MahiNet uses a CNN, i.e., ResNet by He et al. (2016), as a backbone network to extract features from raw images. The CNN feeds features into coarse-class and fine-class classifiers, and the results are combined to produce the final prediction according to fine classes as probabilities. In this way, both the coarse-class and the fine-class classifiers mutually help each other within MahiNet: the former helps to narrow down the candidates for the latter, while the latter provides multiple attributes per coarse class that can regularize the former. This design leverages the relationship between fine classes, and mitigates the difficulty caused by "many class" problem. To the best of our knowledge, we are the first to successfully employ the class hierarchy information to improve few-shot learning. Previous works (Ren et al., 2018) cannot achieve improvement after using the same information.

To address the "few-shot" problem, we apply two types of classifiers in MahiNet, i.e., MLP and K-nearest neighbor (KNN), which have advantages in many-shot and few-shot situations, respectively. We always use MLP for coarse classification, and KNN for fine classification. With a sufficient amount of data in supervised learning, MLP is combined with KNN for fine classification; and in

meta-learning when less data is available, we also use KNN for coarse classification to assist MLP. In Table 1, we provide a brief comparison of MahiNet with other popular models on the learning scenarios they excel.

To make the KNN learnable and more adaptive to classes with few-shot data, we use an attention module to learn the similarity/distance metric used in KNN, and a re-writable memory of limited size to store and update KNN support set during training. In supervised learning, it is necessary to maintain and update a relatively small memory (7.2% of the dataset) by selecting a few samples, because conducting a KNN search over all available training samples is too computationally expensive in computation. In meta-learning, the attention module can be treated as a meta-learner that learns a universal similarity metric for different tasks.

We extract a large subset of ImageNet (Deng et al., 2009) "*mcfs*ImageNet" as a benchmark dataset specifically designed for MCFS problem. It contains 139,346 images from 77 non-overlapping coarse classes composed of 754 randomly sampled fine classes, each has only $\sim 180$ images. Imbalance between the different classes are preserved to reflect the imbalance in practical problems. We further extract "*mcfs*Omniglot" from Omniglot (Lake et al., 2011) for the same purpose. We will make them publicly available later. In experiments on these two datasets, MahiNet outperforms the widely used ResNet (He et al., 2016) in supervised learning. In meta-learning scenario where each task convers many classes, it shows more promising performance than popular few-shot methods including prototypical networks (Snell et al., 2017) and relation networks (Yang et al., 2018).

## 2 MEMORY-AUGMENTED HIERARCHICAL-CLASSIFICATION NETWORK

### 2.1 PROBLEM FORMULATION

We study supervised learning and meta-learning given a training set of $n$ samples $\mathbb{D} = \{(x_i, y_i, z_i)\}_{i=1}^n$, where each sample $x_i \in \mathcal{X}$ is associated with a fine-class label $y_i \in \mathcal{Y}$ and a coarse-class label $z_i \in \mathcal{Z}$, and is sampled from a data distribution $\mathcal{D}$, i.e., $(x_i, y_i, z_i) \sim \mathcal{D}$. Here, $\mathcal{Y}$ denotes the set of all the fine classes, and $\mathcal{Z}$ denotes the set of all the coarse classes. To define a class hierarchy for $\mathcal{Y}$ and $\mathcal{Z}$, we further assume that each coarse class $z \in \mathcal{Z}$ covers a subset of fine classes $\mathcal{Y}_z$, and that distinct coarse classes are associated with disjoint subsets of fine classes, i.e., for any $z_1$, $z_2 \in \mathcal{Z}$, we have $\mathcal{Y}_{z_1} \cap \mathcal{Y}_{z_2} = \emptyset$. Our goal is fine-class classification by using the class hierarchy information. In particular, the supervised learning in this case can be formulated as:

$$\min_{\Theta} \mathbb{E}_{(x,y,z)\sim\mathcal{D}} - \log \Pr(y|x; \Theta), \tag{1}$$

where $\Theta$ is the model parameters. In practice, we solve the corresponding empirical risk minimization (ERM) during training, i.e.,

$$\min_{\Theta} \sum_{i=1}^n - \log \Pr(y_i|x_i; \Theta). \tag{2}$$

In contrast, meta-learning aims to maximize the expectation of the prediction likelihood of a task drawn from a distribution of tasks. Specifically, we assume that the subset of fine classes $T$ for each task is sampled from a distribution $\mathcal{T}$, and the problem is formulated as

$$\min_{\Theta} \mathbb{E}_{T\sim\mathcal{T}} \left[ \mathbb{E}_{(x,y,z)\sim\mathcal{D}_T} - \log \Pr(y|x; \Theta) \right], \tag{3}$$

where $\mathcal{D}_T$ refers to the distribution of samples with label $y_i \in T$. The corresponding ERM is

$$\min_{\Theta} \sum_{T} \left[ \sum_{i\in\mathbb{D}_T} - \log \Pr(y_i|x_i; \Theta) \right], \tag{4}$$

where $T$ is a task (defined by a subset of fine classes) sampled from distribution $\mathcal{T}$, and $\mathbb{D}_T$ is a training set sampled from $\mathcal{D}_T$.

To leverage the coarse class information of $z$, we write $\Pr(y|x; \Theta)$ in Eq. (1) and Eq. (3) as

$$\Pr(y|x; \Theta) = \sum_{z\in\mathcal{Z}} \Pr(y|z, x; \Theta_f) \Pr(z|x; \Theta_c), \tag{5}$$

where $\Theta_f$ and $\Theta_c$ are the model parameters for fine classifier and coarse classifier, respectively[1]. Accordingly, given a specific sample $(x_i, y_i, z_i)$ with its ground truth labels for coarse and fine

---

[1]For simplicity, we neglect model parameters $\theta^{CNN}$ for feature extraction here.

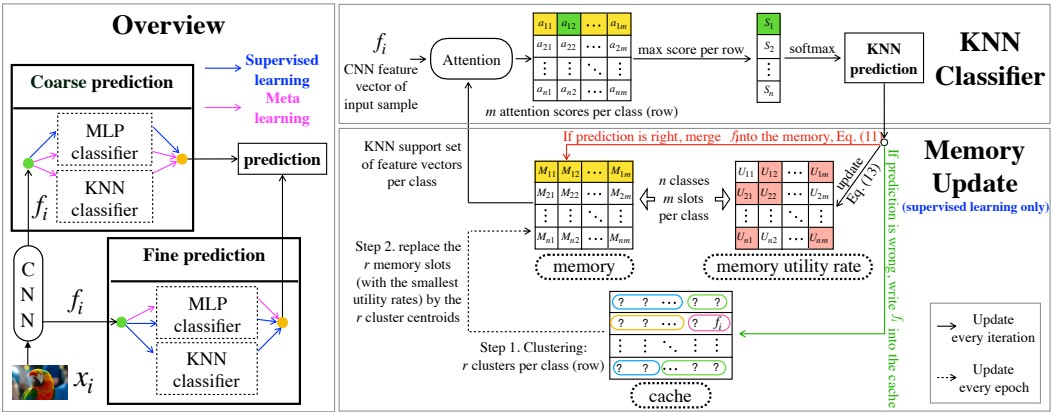

Figure 2: **Left:** MahiNet. The final fine-class prediction combines predictions based on both fine classes and coarse classes, each of which is produced by an MLP classifier or/and an attention-based KNN classifier. **Top right:** KNN classifier with learnable similarity metric and updatable support set. Attention provides a similarity metric $a_{j,k}$ between each input sample $f_i$ and a small support set per class stored in memory $M_{j,k}$. The learning of KNN classifier aims to optimize 1) the similarity metric parameterized by the attention, detailed in Sec. 2.3; and 2) a small support set of feature vectors per class stored in memory, detailed in Sec. 2.4. **Bottom right:** The memory update mechanism. In meta-learning, the memory stores the features of all training samples of a task. In supervised learning, the memory is updated during training as follows: for each sample $x_i$ within an epoch, if the KNN classifier produces correct prediction, $f_i$ will be merged into the memory; otherwise, $f_i$ will be written into a "cache". At the end of each epoch, we apply clustering to the samples per class stored in the cache, and use the resultant centroids to replace $r$ slots of the memory with the smallest utility rate.

classes, we can write $\Pr(y_i|x_i; \Theta)$ in Eq. (2) and Eq. (4) as follows.

$$\Pr(y_i|x_i; \Theta) = \Pr(y_i|z_i, x_i; \Theta_f) \Pr(z_i|x_i; \Theta_c). \qquad (6)$$

Suppose that a DNN model already produces a logit $a_y$ for each fine class $y$, and a logit $b_z$ for each coarse class $z$, the two probabilities in the right hand side of Eq. (6) are computed by applying softmax function to the logit values in the following way.

$$\Pr(y_i|z_i, x_i; \Theta_f) = \frac{\exp(a_{y_i})}{\sum_{y \in \mathcal{Y}_{z_i}} \exp(a_y)}, \quad \Pr(z_i|x_i; \Theta_c) = \frac{\exp(b_{z_i})}{\sum_{z \in \mathcal{Z}} \exp(a_z)}. \qquad (7)$$

Therefore, we integrate both the fine-class label and coarse-class label in an ERM, whose goal is to maximize the likelihood of the ground truth fine-class label. Given a DNN that can produce two logit vectors $a$ and $b$ for fine class and coarse class, we can train it for supervised learning or meta-learning by solving the ERM problems in Eq. (2) or Eq. (4) (with Eq. (6) and Eq. (7) plugged in).

## 2.2 NETWORK ARCHITECTURE

To address MCFS problem in both supervised learning and meta-learning scenarios, we developed a universal model, MahiNet, as in Figure 2. MahiNet uses a CNN to extract features from raw inputs, and then applies two modules to produce coarse-class prediction and fine-class prediction, respectively. Each module includes one or two classifiers: either an MLP or an attention-based KNN classifier or both. Intuitively, MLP performs better when data is sufficient, while the KNN classifier is more stable in few-shot scenario. Hence, we always apply MLP to coarse prediction and apply KNN to fine prediction. In addition, we use KNN to assist MLP for coarse module in meta-learning, and use MLP to assist KNN for fine module in supervised learning. In the attention-based KNN classifier, an attention module is trained to compute the similarity between two samples, and a re-writable memory is maintained with a highly representative support set for KNN prediction.

Our method for learning a KNN classifier combines the ideas from two popular meta-learning methods, i.e., matching networks (Vinyals et al., 2016) that aims to learn a similarity metric, and prototypical networks (Snell et al., 2017) that aims to find a representative center per class for NN search. However, our method relies on an augmented memory rather than a bidirectional RNN for retrieving of NN in matching networks. In contrast to prototypical networks, that only have one prototype per class, we allow multiple prototypes as long as they can fit in the memory budget. Together these two mechanisms prevent the confusion caused by subtle differences between classes in many-class scenario. Notably, MahiNet can also be extended to "life-long learning" given this memory updat-

---

**Algorithm 1** Training of MahiNet for Supervised Learning

---

**Input:** Training set $\mathbb{D} = \{(\boldsymbol{x}_i, y_i, z_i)\}_{i=1}^n$;
    Randomly initialized $\theta_f^{KNN}$, pre-trained $\theta^{CNN}$, $\theta_c^{MLP}$ and $\theta_f^{MLP}$;
    Hyper-parameters: memory update parameters $r$, $\gamma$, $\mu$ and $\eta$; learning rate and its scheduler;
1: **while** no converge **do**
2:   **for** mini-batch $\{(\boldsymbol{x}_i, y_i, z_i)\}_{i \in B}$ in $\mathbb{D}$ **do**
3:    Compute fine-class logits $a$ and coarse-class logits $b$ from the outputs of MLP/KNN classifiers;
4:    Apply one step of mini-batch SGD for ERM in Eq. (2) (with Eq. (6) and Eq. (7) plugged in);
5:    **for** sample in the mini-batch **do**
6:     Update the memory $\boldsymbol{M}$ according to Eq. (11);
7:     Update the utility rate $\boldsymbol{U}$ according to Eq. (13);
8:     Expand the feature cache $\mathbb{C}$ according to Eq. (12);
9:    **end for**
10:   **end for**
11:   **for** each fine class $j$ in $\mathcal{Y}$ **do**
12:    Fine the indexes of the $r$ smallest values in $\boldsymbol{U}_j$, denoted as $\{k_1, k_2, ..., k_r\}$;
13:    Clustering of the feature vectors within cache $\mathbb{C}_j$ to $r$ clusters with centroids $\{c_1, c_2, ..., c_r\}$;
14:    Replace the $r$ memory slots by centroids: $\boldsymbol{M}_{j,k_i} = c_i$ for $i \in [r]$;
15:   **end for**
16: **end while**

---

ing mechanism. We do not adopt the architecture used in Mishra et al. (2018) since it requires the representations of all historical data to be stored.

## 2.3 LEARNING A KNN SIMILARITY METRIC WITH AN ATTENTION MODULE

In MahiNet, we train an attention module to compute the similarity used in the KNN classifier. The attention module learns a distance metric between the feature vector $f_i$ of a given sample $\boldsymbol{x}_i$ and any feature vector from the support set stored in the memory. Specifically, we use the dot product attention similar to the one adopted in Vaswani et al. (2017) for supervised learning, and use an Euclidean distance based attention for meta-learning, following the instruction from Snell et al. (2017). Given a sample $\boldsymbol{x}_i$, we compute a feature vector $\boldsymbol{f}_i \in \mathbb{R}^d$ by applying a backbone CNN to $\boldsymbol{x}_i$. In the memory, we maintain a support set of $m$ feature vectors for each class, i.e., $\boldsymbol{M} \in \mathbb{R}^{C \times m \times d}$, where $C$ is the number of classes. The KNN classifier produces the class probabilities of $\boldsymbol{x}_i$ by first calculating the attention scores between $\boldsymbol{f}_i$ and each feature vector in the memory, as follows.

$$a(\boldsymbol{f}_i, \boldsymbol{M}_{j,k}) = \frac{g(\boldsymbol{f}_i) \cdot h(\boldsymbol{M}_{j,k})}{\|g(\boldsymbol{f}_i)\| \|h(\boldsymbol{M}_{j,k})\|} \ \ \text{or} \ -\|g(\boldsymbol{f}_i) - h(\boldsymbol{M}_{j,k})\|_2, \ \ \forall j \in [C], \ k \in [m], \quad (8)$$

where $g$ and $h$ are learnable transformations for $\boldsymbol{f}_i$ and the feature vectors in the memory. We select the $K$ nearest neighbors of $\boldsymbol{f}_i$ among the $m$ feature vectors for each class $j$, and compute the sum of their similarity scores as the attention score of $\boldsymbol{f}_i$ to class $j$, i.e.,

$$s(\boldsymbol{f}_i, \boldsymbol{M}_j) = \max_{N \subseteq [m], |N| \leq K} \sum_{k \in N} a(\boldsymbol{f}_i, \boldsymbol{M}_{j,k}), \ \ \forall j \in [C]. \quad (9)$$

We usually find $K = 1$ is sufficient in practice. The predicted class probability is derived by applying a softmax function to the attention scores of $\boldsymbol{f}_i$ over all $C$ classes, i.e.,

$$\Pr(y_i = j) \triangleq \frac{\exp\left(s(\boldsymbol{f}_i, \boldsymbol{M}_j)\right)}{\sum_{j'=1}^{C} \exp\left(s(\boldsymbol{f}_i, \boldsymbol{M}_{j'})\right)}, \ \ \forall j \in [C]. \quad (10)$$

## 2.4 MEMORY MECHANISM UPDATING SUPPORT SET OF KNN

Ideally, the memory $\boldsymbol{M} \in \mathbb{R}^{C \times m \times d}$ can store all available training samples as the support set of the KNN classifier. In meta learning, in each episode, we sample a task with $C$ classes and $m$ training samples per class, and store them in the memory. Due to the small amount of training data for each task, we can store all data in the memory. In supervised learning, we only focus on one task, which is possible to have a large training set that cannot be entirely stored in the memory. Hence, we set up a budget hyper-parameter $m$ for each class. $m$ is the maximal number of feature vectors to be stored for one class. Moreover, we develop a memory update mechanism to maintain a small memory with diverse and representative feature vectors (t-SNE visualization can be found in Figure 4 in Appendix E). Intuitively, it can choose to forget or merge feature vectors that are no longer representative, and select new important feature vectors into memory.

---

**Algorithm 2** Training of MahiNet for Meta-Learning

---

**Input:** Training set $\mathbb{D} = \{(\boldsymbol{x}_i, y_i, z_i)\}_{i=1}^n$ and fine class set $\mathcal{Y}$;
        Parameters: randomly initialized $\theta^{CNN}, \theta_c^{MLP}, \theta_f^{MLP}$, and $\theta_f^{KNN}$;
        Hyper-parameters: learning rate, scheduler; for each class, number of queries $n_s$, support set size $n_S$;
1: **while** not converge **do**
2:      Sample a task $T \sim \mathcal{T}$ as a subset of fine classes $T \subseteq \mathcal{Y}$.
3:      **for** class $j$ in $T$ **do**
4:         Randomly sample $n_s$ data points of class $j$ from $\mathbb{D}$ to be the support set $\mathbb{S}_j$ of class $j$.
5:         Randomly sample $n_q$ data points of class $j$ from $\mathbb{D} \backslash \mathbb{S}_j$ to be the query set $\mathbb{Q}_j$ of class $j$.
6:      **end for**
7:      **for** mini-batch from $\mathbb{Q}$ **do**
8:         Compute fine-class logits $a$ and coarse-class logits $b$ from the outputs of MLP/KNN classifiers;
9:         Apply one step of mini-batch SGD for ERM in Eq. (4) (with Eq. (6) and Eq. (7) plugged in);
10:      **end for**
11: **end while**

---

We will show later in experiments that a small memory can result in sufficient improvement, while the time cost of memory updating is negligible. During training, for the data that can be correctly predicted by the KNN classifier, we merge its feature with corresponding slots in the memory by computing their convex combination, i.e.,

$$\boldsymbol{M}_{j,k} = \{ \begin{array}{ll} \gamma \times \boldsymbol{M}_{j,k} + (1-\gamma) \times \boldsymbol{f}_i, & \text{if } \hat{y}_i = y_i \\ \boldsymbol{M}_{j,k}, & \text{otherwise} \end{array} , \tag{11}$$

where $y_i$ is the ground truth label, and $\gamma = 0.95$ is a combination weight that works well in most of our empirical studies; for input feature vector that cannot be correctly predicted, we write it to a cache $\mathbb{C} = \{\mathbb{C}_1, ..., \mathbb{C}_C\}$ that stores the candidates written into the memory for the next epoch, i.e.,

$$\mathbb{C}_j = \{ \begin{array}{ll} \mathbb{C}_j, & \text{if } \hat{y}_i = y_i \\ \mathbb{C}_j \cup \{\boldsymbol{f}_i\}, & \text{otherwise} \end{array} , \tag{12}$$

Concurrently, we record the utility rate of the feature vectors in the memory, i.e., how many times each feature vector being selected into the $K$ nearest neighbor during the epoch. The rates are stored in a matrix $\boldsymbol{U} \in \mathbb{R}^{C \times m}$, and we update it as follows.

$$\boldsymbol{U}_{j,k} = \{ \begin{array}{ll} \boldsymbol{U}_{j,k} \times \mu, & \text{if } \hat{y}_i = y_i \\ \boldsymbol{U}_{j,k} \times \eta, & \text{otherwise} \end{array} , \tag{13}$$

where $\mu \in (1, 2)$ and $\eta \in (0, 1)$ are hyper-parameters.

At the end of each epoch, we cluster the feature vectors per class in the cache, and obtain $r$ cluster centroids as the candidates for the memory update in the next epoch. Then, for each class, we replace $r$ feature vectors in the memory that have the smallest utility rate with the $r$ cluster centroids.

## 3    TRAINING STRATEGIES

As shown in the network structure in Figure 2, in supervised learning and meta learning, we use different combinations of MLP and KNN to produce fine-class and coarse-class predictions. The classifiers are combined by summing up their logits for each class, and a softmax function is used to generate the class probabilities. Assume the MLP classifiers for the coarse classes and the fine classes are $\phi(\cdot; \theta_c^{MLP})$ and $\phi(\cdot; \theta_f^{MLP})$, the KNN classifiers for the coarse classes and the fine classes are $\phi(\cdot; \theta_c^{KNN})$ and $\phi(\cdot; \theta_f^{KNN})$. In supervised learning, the model parameters are $\theta^{CNN}$, $\Theta_c = \theta_c^{MLP}$ and $\Theta_f = \{\theta_f^{MLP}, \theta_f^{KNN}\}$; in meta-learning setting, the model parameters are $\theta^{CNN}$, $\Theta_c = \{\theta_c^{MLP}, \theta_c^{KNN}\}$ and $\Theta_f = \theta_f^{KNN}$.

According to Sec. 2.1, we train MahiNet for supervised learning by solving the ERM problems in Eq. (2) and by solving Eq. (4) for meta-learning. As previously mentioned, the logits (for either fine classes or coarse classes) used in those ERM problems are obtained by summing up the logits produced by the corresponding combination of classifiers.

**Training MahiNet for Supervised learning.** In supervised learning, the memory update relies heavily on the clustering of the merged feature vectors in the cache. To achieve relatively high-quality feature vectors, We first pretrain the CNN+MLP model by using the standard backpropagation to minimize the sum of cross entropy loss on both coarse-classes and fine-classes and then fine-tune the whole model (including the fine-class KNN classifier) with memory updates. The training procedure of the fine-tune stage is explained in Alg. 1.

Table 2: Comparison of the statistics for *mcfs*ImageNet and previously used datasets. "#c" and "#f" denote the number of coarse classes and fine classes, respectively. "-" means "not applicable".

| | Meta Learning | | | | | | Supervised Learning | | | | #image | image size |
|---|---|---|---|---|---|---|---|---|---|---|---|---|
| | Train | | Val | | Test | | Train | | Test | | | |
| | #c | #f | #c | #f | #c | #f | #c | #f | #c | #f | | |
| ImageNet-1k | - | - | - | - | - | - | 1 | 1000 | 1 | 1000 | 1.43M | 224 |
| *mini*ImageNet | 1 | 64 | 1 | 16 | 1 | 20 | - | - | - | - | 0.06M | 84 |
| Omniglot | 33 | 1028 | 5 | 172 | 13 | 423 | - | - | - | - | 0.03M | 28 |
| *mcfs*Omniglot | 50 | 973 | 50 | 244 | 50 | 1624 | - | - | - | - | 0.03M | 28 |
| **mcfsImageNet** | 77 | 482 | 61 | 120 | 68 | 152 | 77 | 754 | 77 | 754 | 0.14M | 112 |

**Training MahiNet for Meta-learning.** In meta learning, the memory is constant and stores features extracted from the support set for KNN classifier. The detailed training procedure can be found in Alg. 2. In summary, we sample each training task by randomly sampling a subset of fine classes, and then randomly sample a support set $\mathbb{S}$ and a query set $\mathbb{Q}$. We store the CNN feature vectors of $\mathbb{S}$ in the memory, and train MahiNet to produce correct predictions for the samples in $\mathbb{Q}$. When sampling the training/test tasks, we allow new fine classes that were not covered in any training task to appear as test tasks, but the ground set of the coarse classes is fixed for both training and test. Hence, every coarse class appearing in any test task has been seen in previous training, but the corresponding fine classes belonging to this coarse class in training and test tasks can vary.

# 4 EXPERIMENTS

We propose two benchmark datasets specifically for MCFS Problem: *mcfs*ImageNet & *mcfs*Omniglot, and compare them with several existing datasets in Table 2. Please see more details in Appendix A. Our following experimental study focuses on these two datasets.

## 4.1 SUPERVISED LEARNING EXPERIMENTS

**Experiments on *mcfs*ImageNet.** We use ResNet18 for the backbone CNN. The transformations $g$ and $h$ in the attention module are two fully connected layers followed by group normalization (Wu & He, 2018) with a residual connection. See more detailed parameter choices in Appendix B.

Table 3 compares MahiNet with the supervised learning model (i.e., ResNet18) and meta learning model (i.e., prototypical networks). The results show that MahiNet outperforms the specialized models, such as ResNet18 in MCFS scenario. Prototypical Net is a meta-learning model designed to solve few-shot classification problems. We train it in a supervised learning manner (i.e., on a single task with many classes and relatively much more samples per class),

Table 3: Different models' performance of supervised learning (test accuracy) on *mcfs*ImageNet.

| Model | Hierarchy | Accuracy |
|---|---|---|
| Prototypical Net | N | 2.7% |
| ResNet18 | N | 48.6% |
| MahiNet w/o KNN | Y | 49.1% |
| MahiNet | Y | **49.9%** |

and include it in the comparison to test its performance on MCFS problem. Prototypical network fails to solve MCFS problem in the supervised learning scenario. To separately measure the contribution of the class hierarchy and the attention-based KNN classifier, we conduct an ablation study that removes the KNN classifier from MahiNet. The results show that MahiNet outperforms ResNet18 even when only using the extra coarse-label information during training, and that using a KNN classifier further improves the performance. For each epoch, the average clustering time is 30s and is only 7.6% of the total epoch time (393s). Within an epoch, the memory update time (0.02s) is only 9% of the total iteration time (0.22s).

## 4.2 META-LEARNING EXPERIMENTS

**Experiments on *mcfs*ImageNet.** We use the same backbone CNN, $g$, and $h$ as in supervised learning. In each task, we sample the same number of classes for training and test, and follow the training procedure in Snell et al. (2017). More detailed parameters can be found in Appendix B.

Table 4: Comparison *w.r.t.* the accuracy (%) of different approaches in meta-learning scenario on *mcfs*ImageNet. Test accuracy is reported as the averaged over 600 test episodes along with the corresponding 95% confidence intervals are reported. In the first row, "n-k" represents $n$-way (class) $k$-shot. Mem-1, Mem-2, and Mem-3 indicate 3 different kinds of memory. In 50-way experiments, Relation Net stops to improve after the first few iterations and fails to achieve comparable performance (more details in Appendix D).

| Model | Hierarchy | 5-10 | 20-5 | 20-10 | 50-5 | 50-10 |
|---|---|---|---|---|---|---|
| ResNet18 (He et al., 2016) | N | 60.7 | 58.6 | 67.2 | 48.9 | 56.8 |
| Prototypical Net (Snell et al., 2017) | N | 78.48±0.66 | 67.78±0.37 | 70.11±0.38 | 57.74±0.24 | 62.12±0.24 |
| Relation Net (Yang et al., 2018) | N | 74.12±0.78 | 52.66±0.43 | 55.45±0.46 | N/A | N/A |
| MahiNet (Mem-1) w/o Attention & Hierarchy | N | 79.04±0.67 | 68.46±0.38 | 71.13±0.38 | 58.09±0.24 | 62.18±0.22 |
| MahiNet (Mem-2) w/o Attention & Hierarchy | N | 77.41±0.71 | 66.89±0.40 | 71.72±0.37 | 55.25±0.23 | 59.38±0.23 |
| MahiNet (Mem-2) w/o Hierarchy | N | 76.85±0.67 | 66.43±0.41 | 70.01±0.38 | 55.13±0.23 | 59.22±0.23 |
| MahiNet (Mem-3) w/o Hierarchy | N | 78.27±0.68 | 67.03±0.41 | 71.20±0.37 | 57.98±0.24 | 62.40±0.23 |
| MahiNet (Mem-1) | Y | 80.64±0.64 | 68.99±0.40 | 72.78±0.37 | 58.56±0.25 | 62.70±0.24 |
| MahiNet (Mem-3) | Y | **80.74**±0.66 | **70.11**±0.41 | **73.50**±0.36 | **58.80**±0.24 | **62.80**±0.24 |

Table 5: Comparison on *mcfs*Omniglot. Symbols and settings are the same as Table 4.

| Model | Hierarchy | 5-5 | 20-5 | 50-5 | 100-5 |
|---|---|---|---|---|---|
| Prototypical Net (Snell et al., 2017) | N | 99.10±0.15 | 98.84±0.11 | 97.94±0.08 | 96.67±0.09 |
| Mahinet (Mem-1) w/o Hierarchy | N | 99.17±0.16 | 98.82±0.11 | 97.96±0.09 | 96.62±0.09 |
| Mahinet (Mem-3) w/o Hierarchy | N | 99.31±0.18 | 98.89±0.11 | 97.93±0.09 | 96.58±0.09 |
| MahiNet(Mem-3) | Y | **99.40**±0.15 | **99.00**±0.16 | **98.10**±0.17 | **96.70**±0.17 |

Table 4 shows that MahiNet outperforms the supervised learning baseline (ResNet18) and the meta-learning baseline (Prototypical Net). For ResNet18, we follow the fine-tune baseline in Finn et al. (2017). To evaluate the contributions of each component in MahiNet, we show results of several variants in Table 4. "Attention" indicates parametric functions for $g$ and $h$, otherwise using identity mapping. "Hierarchy" indicates the assist of class hierarchy. For a specific task, "Mem-1" stores the average feature of all training samples for each class; "Mem-2" stores all features of the training samples; "Mem-3" is the union of "Mem-1" and "Mem-2". Table 4 implies: (1) Class hierarchy information can incur steady performance across all tasks; (2) Combining "Mem-1" and "Mem-2" outperforms using either of them independently; (3) Attention should be learned with class hierarchy in MCFS problem. Because the data is usually insufficient to train a reliable similarity metric to distinguish all fine classes, but distinguishing the fine classes in each coarse class is much easier.

**Experiments on *mcfs*Omniglot**. We conduct experiments on the secondary benchmark *mcfs*ImageNet. We use the same training setting as for *mcfs*ImageNet. Following Santoro et al. (2016), *mcfs*Omniglot is augmented with rotations by multiples of 90 degrees. We do not use ResNet18 on *mcfs*Omniglot, since *mcfs*Omniglot is a $28 \times 28$ small dataset, which would be easy for ResNet18 to overfit. Therefore, we use four consecutive convolutional layers as the backbone CNN and compare MahiNet with prototypical networks as in Table 5. We do ablation study on MahiNet with/without hierarchy and MahiNet with different kinds of memory. "Mem-3", i.e., the union of "Mem-1" and "Mem-2", outperforms "Mem-1", and "Attention" mechanism can improve the performance. Additionally, MahiNet outperforms other compared methods, which indicates the class hierarchy assists to make more accurate predictions. In summary, experiments on the small-scale and large-scale datasets show that class hierarchy brings a stable improvement.

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

## A  Two New Benchmarks for MCFS Problem: *mcfs*ImageNet & *mcfs*Omniglot

ImageNet (Deng et al., 2009) may be the most widely used large-scale benchmark dataset for image classification. However, although it provides hierarchical information about class labels, it cannot be directly used to test the performance of MCFS learning methods. Because a fine class may belong to multiple coarse classes, and in MCFS problem, each sample has only one unique coarse-class label. In addition, ImageNet does not satisfy the criteria of "few-shot" per class. *mini*ImageNet (Vinyals et al., 2016) has been widely used as a benchmark dataset in meta-learning community to test the performance on few-shot learning task. *mini*ImageNet is a subset of data extracted from ImageNet; however, its data are sourced from only 80 fine classes, which is not "many-class" nor does this carry a class hierarchy.

Hence, to develop a benchmark dataset specifically for the purpose of testing the performance of MCFS learning methods, we extracted a subset of images from ImageNet and created a dataset called "*mcfs*ImageNet". Table 2 compares the statistics of *mcfs*ImageNet with several benchmark datasets, and more details of the class hierarchies in mcfsImageNet are given in the Appendix F. Comparing to the original ImageNet, we avoided selecting the samples that belong to more than one coarse classes into *mcfs*ImageNet to meet the class hierarchy requirements of MCFS problem, i.e., each fine class only belongs to one coarse class. Compared to *mini*ImageNet, *mcfs*ImageNet is about $5\times$ larger, and covers 754 fine classes - many more than the 80 fine classes in *mini*ImageNet. Moreover, on average, each fine class only has $\sim 185$ images for training and test, which is typical MCFS scenarios. Additionally, the number of coarse classes in *mcfs*ImageNet is 77, which is many less than 754 of the fine classes. This is consistent with the data properties found in many practical applications, where the coarse-class labels can only provide weak supervision, but each coarse class has sufficient training samples. Further, we avoided selecting coarse classes which were too broad or contained too many very different fine classes. For example, the "Misc" class in ImageNet has 20400 sub-classes, and includes both animal (3998 sub-classes) and plant (4486 sub-classes).

Omniglot (Lake et al., 2011) is a small hand-written character dataset with two levels. However, new coarse classes appear in the test set, which is inconsistent with our MCFS settings (all the coarse classes are exposed in training, but new fine classes can appear during test). As a result, we re-split Omniglot to fulfill the MCFS problem requirement and the class hierarchy information are listed in Appendix G.

## B  Experimental Setup

**Setup for the supervised learning.** We use ResNet18 (He et al., 2016) for the backbone CNN. The transformation functions $g$ and $h$ in the attention module are two fully connected layers followed by group normalization (Wu & He, 2018) with a residual connection. We set the memory size to $m = 12$ and the number of clusters to $r = 3$, which can achieve a better trade-off between memory cost and performance. Batch normalization (Ioffe & Szegedy, 2015) is applied after each convolution and before activation. During pre-training, we apply the cross entropy loss on the probability predictions in Eq. (7). During fine-tuning, we fix the $\theta^{CNN}$, $\theta_c^{MLP}$, and $\theta_f^{MLP}$ to ensure the fine-tuning process is stable. We use SGD with a mini-batch size of 128 and a cosine learning rate scheduler with an initial learning rate 0.1. $\mu = 1.05$, $\eta = 0.95$, a weight decay of 0.0001, and a momentum of 0.9 are used. We train the model for 100 epochs during pre-training and 90 epochs for the fine-tuning.

**Setup for the meta learning.** We use the same backbone CNN, $g$, and $h$ as in supervised learning. In each task, we sample the same number of classes for training and test, and follow the training procedure in Snell et al. (2017). We set an initial learning rate to $10^{-3}$ and reduce it by a factor $2\times$ every 10k iterations. Our model is trained by Adam (Kingma & Ba, 2015) with a mini-batch size of 128, a weight decay of 0.0001, and a momentum of 0.9. We train the model for 25k iterations in total. For class hierarchy, the objective function is the sum of the softmax with cross entropy losses on the coarse class and on the fine class, respectively.

## C  RELATED WORKS

Few-shot learning has a long history. Before deep learning, generative models (Fei-Fei et al., 2006) are trained to provide a global prior knowledge for solving the one-shot learning problem. However, with the advent of deep learning techniques, some recent approaches (Wong & Yuille, 2015; Lake et al., 2013) use generative models to encode specific prior knowledge, such as strokes and patches. More recently, Douze et al. (2018) and Wang et al. (2018) have applied hallucinations to training images and to generate more training samples, which converts a few-shot problem to a many-shot problem.

Meta-learning has been used in attempts to solve the few-shot learning problems. Meta learning was first proposed in the last century (Naik & Mammone, 1992; Schmidhuber, 1987), and ) but has recently seen some significant improvements. For example, Lake et al. (2015) proposed a dataset of characters for meta-learningn while Koch et al. (2015) extended this idea into a Siamese network. A more challenging dataset (Ravi & Larochelle, 2017; Vinyals et al., 2016) was introduced later. Researchers have also studied RNN and attention based method to overcome the few-shot problem. More recently, Snell et al. (2017) is proposed based on a metric learning equipped KNN. In contrast, Finn et al. (2017) based their approach on the second order optimization. Mishra et al. (2018) uses temporal convolution to address the few-shot image recognition. However, unlike above methods, our model leverages the class hierarchy information, and can be easily applied to both the supervised learning and meta-learning scenarios.

## D  COMPARISON TO VARIANTS OF RELATION NET

**Relation network with class hierarchy.** We train relation network with class hierarchy in the similar manner as in MahiNet. The results are shown in Table 6. It demonstrates that the class hierarchy also improves the accuracy of relation network by more than 1%, which verifies the advantage of using class hierarchy in other models besides MahiNet.

Table 6: The improvement of class hierarchy on relation network.

| Model | Hierarchy | 5 way 5 shot | 5 way 10 shot |
|---|---|---|---|
| Relation Net (Yang et al., 2018) | N | 63.02±0.87 | 74.12±0.78 |
| Relation Net | Y | 66.82±0.86 | 75.31±0.90 |
| MahiNet (Mem-3) | Y | **74.98**±0.75 | **80.74**±0.66 |

**Relation network in high way setting.** For relation network in high way settings, we found that the network is easy to be stuck into a suboptimal solution. After first few iterations, the training loss stays in a high level and the training accuracy stays in a low level. We demonstrate the training loss and training accuracy for the first 100 iterations under different learning rate as Figure 3. The training loss and accuracy keep the same value after 100 iterations.

## E  ANALYSIS OF AUGMENTED MEMORY

**Visualization.** In order to show how representative and diverse the feature vectors selected into memory slots are, we visualize feature vectors in the memory and the rest image feature vectors in t-SNE in Figure 4. In particular, we randomly sample 50 fine classes marked by different colors. Within every class, we show both the selected feature vectors in memory and feature vectors of other images from the same class. It shows that the small number of highly selected feature vectors in memory are diverse and sufficiently representative of the whole class.

**Memory Cost.** In experiments of supervised learning, the memory size required by MahiNet is only $754 \times 12/125321 = 7.2\%$ (12 samples per class for all the 754 fine classes, while the training set includes $125,321$ images in total) of the memory needed to store the whole training set. We also tried to increase the memory size to about $10\%$, but the resultant improvement on performance is negligible. In each task of meta learning, since every class only has few-shot samples, the memory

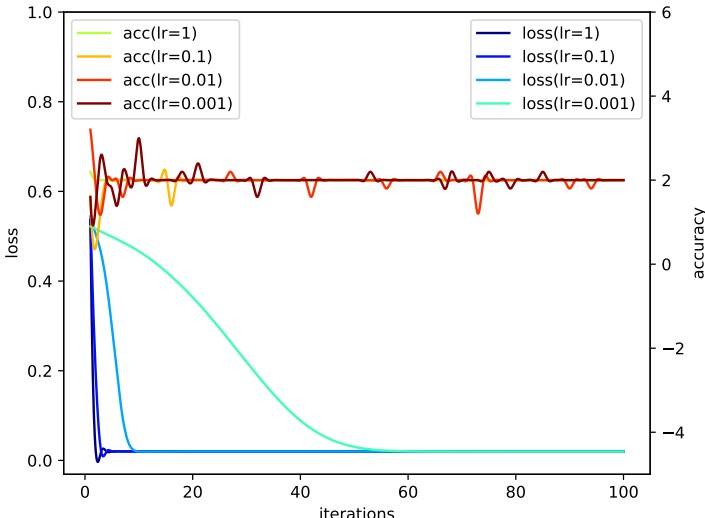

Figure 3: The training loss and accuracy for 50 way 5 shot relation net during the first 100 iterations under different learning rate. Whatever the learning rate is, it quickly converges to a suboptimal point: The training loss: $\approx 0.02$ and the training accuracy: $\approx 2\%$.

required to store all the data is very small. For example, in the 20-way 1-shot setting, the memory only needs to store 20 feature vectors.

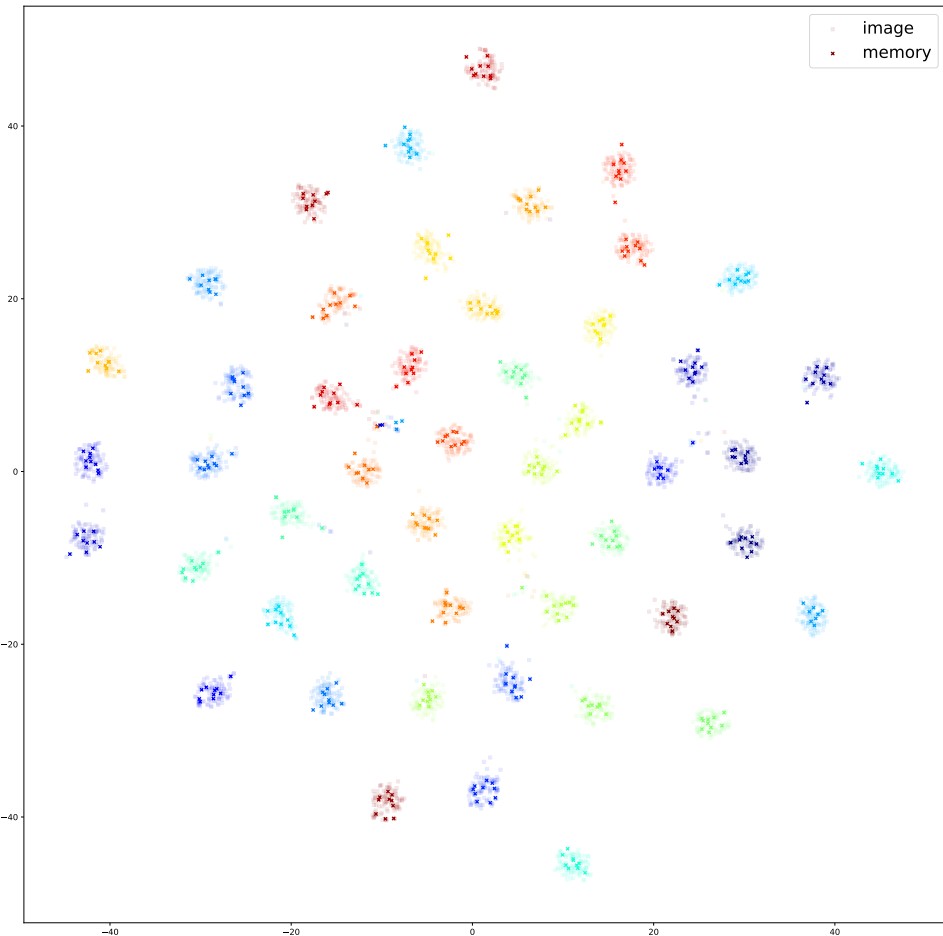

Figure 4: The t-SNE visualization for memory. We randomly sample 50 classes shown as different colors. The image feature and the memory feature share the same color while the image feature has higher transparency.

## F  *mcfs*ImageNet Class Splits

The hierarchy of coarse and fine classes is shown below. Every key (marked in **bold**) in the dictionary is a coarse class, the value is a list of the fine classes of this coarse classes.

**n02470325**: [n02482650, n02481500, n02483362, n02481103, n02482060, n02481235, n02482474, n02481366, n02482286, n02483708]  **n01769347**: [n01782516, n01775370, n01772664, n01773797, n01773549, n01774384, n01777304, n01773157, n01778217, n01770081]  **n12855042**: [n12856479, n12855710, n12855886, n12855494, n12855365, n12856287, n12856680, n12856091]  **n13109733**: [n11705171, n12717224, n12734215, n12491017, n11710136, n11707827, n12634734, n12711984, n12709103, n12753245]  **n07806221**: [n07807834, n07808904, n07806774, n07808268, n07808587, n07806879, n07807472, n07808022, n07807594, n07808479]  **n01428580**: [n02583890, n01447331, n01449374, n02520147, n01439808, n02538562, n01442972, n02543565, n02527057, n01447658]  **n01789386**: [n01793249, n01790812, n01790304, n01795735, n01815036, n01806297, n01793085, n01796729, n01794344, n01797020]  **n12582231**: [n12590499, n12597466, n12587487, n12598027, n12595699, n12596148, n12583855, n12587803, n12597134, n12596345]  **n01816887**: [n01819734, n01819313, n01818299, n01817263, n01819465, n01817346, n01822300, n01821869, n01818832, n01818515]  **n12724942**: [n12728322, n12731029, n12727518, n12729521, n12726159, n12729729, n12728864, n12730544, n12725940, n12728508]  **n07802417**: [n07804543, n07802863, n07889193, n07804900, n07804771, n07804657, n07802963, n07803310]  **n07882497**: [n07628181, n07861247, n07861158, n07594066, n07628068, n07678586, n07594155, n07679034, n07839478, n07678953]  **n01940736**: [n01945143, n01953594, n01944118, n01955084, n01947396, n01959029, n01965529, n01960177, n01963317, n01958346]  **n07911677**: [n07914586, n07915800, n07916437, n07931452, n07918309, n07914995, n07917392, n07918193, n07917133, n07916183]  **n12864545**: [n12865037, n12866002, n12866162, n12866459, n12865824, n12865708, n12865562, n12866635]  **n07886572**: [n07887192, n07890226, n07889990, n07887304, n07887967, n07888816, n07890540, n07890068, n07887461, n07888229]  **n01524359**: [n01561732, n01556182, n01537895, n01561452, n01593553, n01583828, n01539925, n01543632, n01535690, n01538630]  **n12268246**: [n12278865, n12271933, n12278650, n12277150, n12276872, n12270278, n12276477, n12273344, n12271187, n12270741]  **n01674216**: [n01682435, n01682714, n01680983, n01689811, n01687978, n01690466, n01678043, n01680655, n01693175, n01689081]  **n02084071**: [n02088466, n02101006, n02113624, n02091467, n02111277, n02087394, n02102177, n02105412, n02094258, n02106662]  **n02274024**: [n02305085, n02297819, n02304432, n02277742, n02302244, n02303284, n02281015, n02279972, n02276078, n02282257]  **n02441326**: [n02443484, n02445171, n02442572, n02446206, n02445004, n02443346, n02450295, n02448060, n02447762, n02449183]  **n02484322**: [n02488415, n02493793, n02494079, n02493509, n02492660, n02492948, n02487547, n02485536, n02485688, n02490811]  **n07901587**: [n07906111, n07902799, n07905296, n07904865, n07907037, n07904293, n07905386, n07902443, n07903101, n07902336]  **n02801938**: [n02925009, n04024862, n03050864, n03390673, n03883385, n03129848, n02893608, n04204238, n03378342]  **n12387633**: [n12389501, n12388858, n12390099, n12389727, n12390314, n12388989, n12389130, n12388143]  **n04576211**: [n04474035, n03235979, n03435991, n02871314, n03193260, n02740300, n03256166, n03173929, n04026813, n04285965]  **n01661592**: [n01666585, n01664990, n01668665, n01671125, n01669372, n01670802, n01666228, n01669654, n01668892, n01665541]  **n01627424**: [n01636352, n01646555, n01633406, n01632458, n01654637, n01646802, n01641739, n01641930, n01629819, n01631354]  **n02131653**: [n02133704, n02133400, n02132788, n02132466, n02134084, n01322983, n02132580, n02134418]  **n06271778**: [n06272612, n06276501, n06278475, n06273555, n06274760, n06273294, n06273986, n06273414, n06275471, n06277135]  **n13100677**: [n12162425, n12828220, n11731659, n11788727, n12513172, n12828379, n11789589, n12383737, n11769803, n12485981]  **n01726692**: [n01744401, n01738065, n01749939, n01732614, n01737728, n01753180, n01743936, n01733466, n01750167, n01757343]  **n02394477**: [n02419634, n02414290, n02396014, n02412080, n02405440, n02396427, n02415829, n02395931, n02418465, n02427470]  **n01604330**: [n01608814, n01611800, n01624115, n01606809, n01623615, n01606177, n01610100, n01609391, n01609956, n01604968]  **n02373336**: [n02377291, n02391373, n02389026, n02379329, n02378969, n02379081, n02386853, n02379908, n02382204, n02389943]  **n02898711**: [n04311004, n02775897, n03865557, n04479939, n03415486, n03981760, n04108822, n03233744, n03122073, n02986160]  **n01976146**: [n01986214, n01983674, n01988203, n01981276, n01984695, n01980166, n01978455, n01978287, n01982068, n01979874]  **n02062430**: [n02065726, n02071028, n02072798, n02070624, n02065263, n02069412, n02068541, n02071636, n02063662, n02069974]  **n02075927**: [n02080713, n02081798, n02078738, n02077384, n02077787, n02080146, n02080415, n02079851, n02076402, n02078574]  **n13112664**: [n12632335, n11948864,

n12249542, n12861541, n12905135, n12675876, n12921868, n11865874, n12806015, n12482668]
**n07829412**: [n07837002, n07837755, n07837545, n07840520, n07838073, n07831267, n07832416, n07840219, n07835921, n07826091]     **n07892813**: [n07897438, n07898247, n07897975, n07897600, n07899533, n07899769, n07894799, n07894703, n07894451, n07896765]     **n09366017**: [n09398076, n09454744, n09233446, n09435739, n09452291, n09410224, n09245515, n09415671, n09295946, n09344324]     **n01480516**: [n01495493, n01484562, n01492569, n01484285, n01500476, n01485479, n01481498, n01489709, n01496331, n01491006]     **n02974697**: [n02962843, n03031012, n03498441, n03468821, n04570815, n03210372, n04035836, n03986949, n03438863, n04064747]     **n07927197**: [n07928696, n07928998, n07929172, n07927836, n07928163, n07928578, n07928488, n07927716, n07928790, n07928887]     **n11545524**: [n13230843, n13181244, n13193143, n13214485, n13231078, n13229543, n12961879, n13190747, n13183056, n12953712]     **n03206908**: [n03062336, n04130257, n04050933, n04242704, n04263257, n03775546, n02927764, n04499062, n03920288, n02997910]
**n07596684**: [n07609632, n07601686, n07600177, n07643891, n07604956, n07643200, n07608098, n07609215, n07597263, n07608533]     **n07560652**: [n07561590, n07565083, n07562017, n07938007, n07562379, n07565161, n07560903, n07563366, n07564796, n07563207]     **n07583197**: [n07588817, n07588574, n07584423, n07585557, n07588193, n07588111, n07585107, n07586894, n07586318, n07587618]     **n13108841**: [n11646344, n11626585, n11656123, n11626826, n11615607, n11644462, n11615387, n11655974, n11622368, n11662371]     **n02164464**: [n02177196, n02174001, n02173373, n02178411, n02167944, n02175916, n02183096, n02169974, n02180875, n02181724]     **n13085113**: [n11874081, n11898775, n12003167, n12394118, n11949015, n11965218, n11954161, n11920133, n11919761, n12033139]     **n02139199**: [n02146879, n02146371, n02146700, n02140049, n02147173, n02147947, n02147328, n02147591]     **n07907943**: [n07910048, n07910152, n07908812, n07908411, n07910379, n07910970, n07911249, n07908567, n07908647, n07910538]     **n02188699**: [n02197185, n02195526, n02203152, n02201626, n02198859, n02191979, n02196119, n02202124, n02204907, n02192513]     **n07843775**: [n07848936, n07847917, n07848093, n07847827, n07848196, n07849733, n07616046, n07847453, n07849912, n07847585]     **n02121808**: [n02122430, n02123478, n02123045, n02122510, n02123159, n02122298, n02123242, n02122878, n02123394, n02123917]     **n11868814**: [n11876803, n11876204, n11870747, n11875523, n11879722, n11882074, n11882426, n11877646, n11877193, n11876432]     **n02552171**: [n02656301, n02580679, n02555863, n02565324, n01451426, n02663211, n02578928, n02654112, n02572484, n02607470]     **n12997654**: [n13013965, n13075684, n13053608, n13014265, n13013764, n13018088, n13020964, n13011595, n13032618, n13003522]
**n03764276**: [n04006067, n04308273, n03466600, n04457474, n04308397, n03549199, n03811295, n04487894, n04363082, n03466493]     **n09366317**: [n09362945, n09214916, n09230202, n09472597, n09411295, n09421951, n09193705, n09396465, n09269341, n09218641]     **n07679356**: [n07682808, n07685218, n07685399, n07682477, n07683039, n07680761, n07690431, n07684938, n07681691, n07681450]     **n07929519**: [n07919572, n07920872, n07920349, n07919441, n07920222, n07731284, n07919894, n07920663, n07920540, n07921239]     **n02323449**: [n02324514, n02324431, n02327842, n02324587, n02324850, n02327656, n02328150, n02325722, n02326862]     **n13100156**: [n12766869, n11724109, n11734698, n12455950, n11723227, n11773987, n11767877, n12935609, n12767648, n12941220]     **n12334293**: [n12336727, n12338454, n12336224, n12337617, n12316572, n12340581, n12338655, n12340755, n12338146, n12336973]     **n01909422**: [n01914830, n01917882, n01917611, n01917289, n01915700, n01913166, n01916388, n01909906, n01916481]     **n01838038**: [n01839949, n01842235, n01841679, n01840775, n01841441, n01839086, n01840412, n01843065, n01839330, n01840120]     **n02206270**: [n02213663, n02209111, n02213788, n02211627, n02218371, n02221715, n02210427, n02216211, n02221414, n02214499]     **n07712382**: [n07696625, n07696728, n07696839, n07697313, n07698672, n07865105, n07696977, n07697699, n07698401, n07697408]     **n07811416**: [n07935043, n07820960, n07933154, n07816398, n07821758, n07812046, n07818825, n07826340, n07816164, n07820297]     **n12685431**: [n12686077, n12686274, n12686496, n12686676, n12687957, n12686877, n12687044, n12687462, n12687698]     **n12101870**: [n12139921, n12140903, n12140511, n12134486, n12123741, n12126084, n12121610, n12117326, n12130549, n12133462]

## G  *mcfs*Omniglot Class Splits

The hierarchy of coarse and fine classes is shown below. Every key (marked in **bold**) in the dictionary is a coarse class, the value is a list of the fine classes of this coarse classes.

**Training Hierarchy Classes:**

**Mkhedruli (Georgian)**: [character10, character38, character25, character29, character21, character04, character13, character37, character34, character31, character17, character16, character03, character30, character02, character06, character24, character11, character39, character19, character07, character14, character32, character36] **Alphabet of the Magi**: [character18, character05, character17, character03, character08, character01, character15, character07, character09, character10, character19, character11, character20] **Angelic**: [character12, character16, character20, character06, character13, character19, character04, character03, character07, character09, character05] **Early Aramaic**: [character09, character02, character07, character11, character04, character19, character16, character05, character21, character18, character06, character12, character03, character13, character10] **Old Church Slavonic (Cyrillic)**: [character21, character36, character38, character07, character15, character11, character29, character14, character43, character17, character34, character25, character41, character35, character23, character22, character20, character44, character19, character09, character10, character33, character45, character26, character32] **Gujarati**: [character05, character44, character06, character35, character40, character45, character07, character29, character12, character09, character24, character38, character46, character15, character32, character21, character18, character25, character04, character02, character23, character03, character31, character14, character34, character20, character37, character43, character30, character36, character26, character08, character28] **Japanese (katakana)**: [character31, character44, character10, character21, character33, character14, character36, character06, character11, character28, character27, character04, character26, character43, character03, character08, character12, character24, character16, character15, character35, character05, character01, character41, character25, character29, character38, character39, character22, character02] **Syriac (Serto)**: [character14, character23, character03, character04, character18, character06, character02, character22, character15, character09] **Tengwar**: [character21, character19, character15, character13, character03, character11, character16, character10, character18, character05, character14] **Korean**: [character03, character14, character35, character32, character04, character19, character15, character33, character01, character28, character09, character40, character16, character07, character39, character34, character37, character11, character36, character24, character10, character05] **Malayalam**: [character37, character34, character42, character01, character10, character40, character28, character06, character17, character35, character14, character19, character38, character13, character08, character46, character47, character21, character23, character18, character36, character24, character43, character33, character03] **Japanese (hiragana)**: [character44, character05, character13, character19, character31, character40, character24, character50, character07, character11, character06, character01, character32, character10, character46, character04, character47, character45, character27, character42, character36, character09, character34, character49, character12, character28, character37, character26, character23, character52, character38, character17, character25, character30] **Atemayar Qelisayer**: [character06, character08, character09, character16, character05, character14, character11, character01, character10, character13, character02, character20, character26] **Tibetan**: [character31, character30, character21, character18, character20, character34, character29, character10, character23, character16, character42, character17, character15, character03, character33, character01, character25, character13, character09, character05, character40, character26, character38, character06, character28] **Sylheti**: [character03, character28, character17, character13, character23, character27, character08, character22, character05, character14, character12, character16, character04, character18, character20, character24] **Malay (Jawi - Arabic)**: [character27, character11, character12, character13, character23, character36, character26, character33, character31, character39, character32, character40, character21, character09, character14, character17, character28, character38, character06] **Latin**: [character07, character08, character02, character24, character22, character06, character14, character09, character21, character10, character25, character04, character13, character19, character03, character26, character05, character18, character16, character15] **Syriac (Estrangelo)**: [character20, character19, character04, character10, character09, character07, character21, character08, character22, character16, character15, character14, character03, character05, character01, character11, character17] **ULOG**: [character04, character02, character15, character05, character08, character12, character03, character16, character24, character13, character26, character23, character25, character19] **Blackfoot (Canadian Aboriginal Syllabics)**: [character13, character03, character12, character11, character05, character07, character01, character10] **Futurama**: [character22, character20, character08, character11, character05, character06, character16, character02, character13, character03, character14, character15, character17, character12] **Gurmukhi**: [character34, character21, character08, character38, character04, character37, character09, character03, character26, character35, character20, character19, character41, character13, character16, character28, character10, character39, character32, character29, character33, character24, character12, character45, character22, character05] **Ojibwe (Canadian Aboriginal Syllabics)**: [character08, character04, character01, character12, character13, character11, character03, character10, character05, character02] **Greek**: [character13, character14, character06, character22, character01, character19, character08, character02, character09, character07, character04, character18, character12, character03] **Armenian**: [character11, character13, character35, character36, character21, character26, character01, character30, character27,

character32, character40, character05, character25, character16, character22, character19, character39, character15, character14, character33, character09] **Kannada**: [character01, character32, character20, character28, character40, character11, character10, character17, character25, character33, character02, character19, character27, character39, character37, character38, character36, character41, character29, character23, character16, character14, character04, character18, character21, character12, character24, character09] **Manipuri**: [character04, character20, character24, character32, character02, character09, character33, character30, character14, character29, character16, character05, character34, character03, character13, character12, character31, character10, character18, character15, character22, character23] **Oriya**: [character09, character31, character45, character27, character05, character07, character11, character23, character15, character20, character29, character44, character32, character06, character19, character46, character36, character03, character25, character01, character26, character04, character38, character22, character35] **Asomtavruli (Georgian)**: [character10, character12, character17, character11, character03, character24, character25, character35, character32, character22, character21, character29, character14, character23, character34, character20, character33, character26, character16, character28, character04, character05, character08, character06] **Keble**: [character17, character03, character15, character18, character13, character19, character10, character07, character21, character26, character08, character12, character09, character11, character14, character16, character05, character22] **Arcadian**: [character09, character10, character03, character16, character26, character04, character08, character18, character23, character02, character14, character22, character05, character01] **Cyrillic**: [character33, character26, character01, character11, character08, character24, character22, character10, character04, character27, character07, character09, character06, character18, character20, character19, character21, character12, character25, character29, character32] **Hebrew**: [character16, character02, character10, character19, character13, character01, character11, character14, character22, character12, character07, character15, character21, character05] **Avesta**: [character23, character21, character17, character18, character03, character25, character19, character24, character22, character06, character09, character16, character05, character26, character15] **Glagolitic**: [character14, character01, character23, character42, character39, character06, character33, character03, character19, character10, character11, character37, character21, character04, character31, character41, character30, character07, character17, character35, character26, character28, character25, character29] **Ge_ez**: [character18, character11, character07, character22, character21, character04, character06, character13, character15, character25, character20, character12, character08, character05] **Bengali**: [character44, character06, character17, character45, character12, character09, character04, character32, character22, character46, character37, character14, character07, character26, character20, character41, character11, character33, character02, character40, character39, character10, character35, character28, character01, character19, character05, character27, character13] **Atlantean**: [character17, character26, character11, character15, character08, character19, character09, character12, character04, character18, character21, character03, character02, character06, character23] **Sanskrit**: [character20, character40, character24, character01, character37, character39, character28, character42, character13, character09, character02, character25, character23, character33, character19, character07, character35, character36, character17, character11, character38, character15, character16, character31, character03, character10, character26, character21, character29] **Braille**: [character12, character14, character18, character15, character23, character05, character01, character10, character24, character02, character22, character11, character20, character21] **Burmese (Myanmar)**: [character15, character23, character33, character26, character10, character08, character30, character25, character21, character07, character31, character19, character34, character05, character29, character27, character06, character20, character03, character09, character18, character24, character22, character12] **Tagalog**: [character05, character14, character04, character03, character12, character15, character10, character09, character01, character17, character11] **Mongolian**: [character25, character16, character11, character30, character14, character20, character18, character07, character17, character13, character19, character12, character29, character27, character08, character01, character28, character22, character26] **Inuktitut (Canadian Aboriginal Syllabics)**: [character13, character05, character06, character15, character11, character16, character10, character14] **N_Ko**: [character23, character12, character16, character11, character32, character20, character19, character02, character10, character31, character13, character28, character18, character14, character24, character26, character21, character30, character15, character01] **Balinese**: [character19, character06, character17, character13, character11, character01, character24, character02, character04, character20, character23, character12, character14, character18, character09] **Aurek-Besh**: [character23, character20, character05, character17, character14, character06, character15, character09, character22, character13, character07, character03, character19, character16, character24, character08, character10, character02, character12, character21, character18, character11, character25] **Anglo-Saxon_Futhorc**: [character11, character14, character09, character01, character10, character16, character05, character27, character18, character02, character08, character21, character13, character29, character17, character20] **Tifinagh**: [character42, character38, character35, character36, character31, character08, character52, character26, character41, character50, character09, character45, character24, character11, character39,

character19, character34, character05, character21, character16, character23, character37, character04, character28, character03, character32, character10, character43, character40, character48, character17, character44, character49, character06, character20]  **Grantha**: [character39, character13, character12, character33, character35, character15, character16, character27, character20, character19, character30, character26, character42, character22, character23, character03, character02, character07, character38, character40, character25, character18, character08, character37, character32, character24]

## Validation Hierarchy Classes:

**Alphabet_of_the_Magi**: [character16]  **Angelic**: [character01, character10, character11]  **Early_Aramaic**: [character17, character14]  **Oriya**: [character14, character40, character10, character21, character33, character39, character12, character41, character16]  **Gujarati**: [character13, character47, character33, character10, character22]  **Japanese_(katakana)**: [character17, character46, character45, character30, character40, character07, character18]  **Syriac_(Serto)**: [character13, character12, character17, character21, character08, character20, character19]  **Tengwar**: [character09, character06, character22]  **Korean**: [character25, character29, character26, character38, character06]  **Malayalam**: [character12, character09, character30, character11, character31, character44, character39, character32, character04]  **Arcadian**: [character13, character19, character07, character06]  **Atemayar_Qelisayer**: [character18, character24, character25, character22]  **Tibetan**: [character32, character11, character12, character41, character35]  **Sylheti**: [character25, character01, character26, character06]  **Old_Church_Slavonic_(Cyrillic)**: [character16, character28, character01, character12, character40, character42, character39]  **ULOG**: [character07, character22, character06, character17, character14, character20, character18, character10]  **Syriac_(Estrangelo)**: [character12, character06]  **Tagalog**: [character08]  **Blackfoot_(Canadian_Aboriginal_Syllabics)**: [character09, character08]  **Futurama**: [character25, character10, character04, character26, character24, character19, character01]  **Gurmukhi**: [character18, character17, character43, character25, character07, character42, character06, character02]  **Ojibwe_(Canadian_Aboriginal_Syllabics)**: [character06, character09]  **Mkhedruli_(Georgian)**: [character40, character33, character22, character26, character05, character27, character15, character12, character28]  **Armenian**: [character37, character18, character08, character17]  **Kannada**: [character15, character13, character08, character26, character05, character35, character07, character34, character30]  **Manipuri**: [character06, character36, character11, character27, character38, character26]  **Anglo-Saxon_Futhorc**: [character25, character24, character06]  **Asomtavruli_(Georgian)**: [character02, character09, character15, character07, character39, character18, character30]  **Keble**: [character24, character20, character04, character23, character01]  **Japanese_(hiragana)**: [character22, character02, character33, character41, character14, character16, character51]  **Braille**: [character13, character26, character04, character25, character17]  **Hebrew**: [character20, character03, character18, character06, character09]  **Avesta**: [character10, character01, character08, character11, character13, character20, character07]  **Glagolitic**: [character44, character38, character08, character18, character36, character12, character13, character09]  **Latin**: [character23]  **Ge_ez**: [character01, character23, character09, character19]  **Bengali**: [character29, character23, character21, character38]  **Atlantean**: [character13, character20]  **Greek**: [character05, character21, character24, character10]  **Cyrillic**: [character02, character16]  **Burmese_(Myanmar)**: [character14, character13, character32]  **Grantha**: [character17, character14, character31, character21, character01, character11]  **Mongolian**: [character23, character24, character15, character03]  **Inuktitut_(Canadian_Aboriginal_Syllabics)**: [character02, character08, character12, character09]  **N_Ko**: [character33, character03, character09, character06, character25]  **Balinese**: [character15, character07, character03, character05]  **Aurek-Besh**: [character26]  **Tifinagh**: [character30, character15, character54, character33, character02, character18, character07, character13]  **Sanskrit**: [character14, character18, character22, character27, character08]  **Malay_(Jawi_-_Arabic)**: [character08, character05, character02, character16, character19, character18, character29]

## Testing Hierarchy Classes:

**Mkhedruli_(Georgian)**: [character01, character18, character20, character41, character08, character09, character23, character35]  **Alphabet_of_the_Magi**: [character14, character04, character13, character06, character02, character12]  **Angelic**: [character02, character08, character14, character17, character15, character18]  **Early_Aramaic**: [character01, character08, character15, character22, character20]  **Oriya**: [character30, character18, character43, character28, character17, character13, character24, character42, character02, character37, character08, character34]  **Gujarati**: [character17, character39, character19, character11, character01, character16, character41, character27, character48, character42]  **Japanese_(katakana)**: [character20, character32, character42, character09, character13, character37, character19, character34, character23, character47]  **Syriac_(Serto)**: [character05, character07, character10, character11, character01, character16]  **Ge_ez**: [character10, character16, character17, character14, character03, character24, character26, character02]  **Korean**: [character31, character12, character22, character21, character18, character02, character23, character27, char-

acter30, character17, character20, character08, character13] **Malayalam**: [character07, character22, character15, character05, character25, character16, character02, character45, character41, character29, character26, character27, character20] **Atlantean**: [character16, character14, character25, character05, character01, character07, character10, character22, character24] **Atemayar_Qelisayer**: [character19, character21, character12, character04, character07, character17, character15, character23, character03] **Arcadian**: [character24, character20, character25, character12, character21, character17, character15, character11] **Tibetan**: [character04, character36, character19, character37, character07, character39, character22, character02, character24, character08, character27, character14] **Old_Church_Slavonic_(Cyrillic)**: [character02, character13, character08, character06, character30, character27, character03, character05, character24, character31, character37, character18, character04] **Malay_(Jawi_-_Arabic)**: [character04, character03, character15, character37, character20, character34, character24, character35, character25, character30, character07, character10, character01, character22] **Cyrillic**: [character13, character15, character05, character14, character23, character31, character28, character17, character03, character30] **Syriac_(Estrangelo)**: [character23, character18, character13, character02] **Tagalog**: [character07, character06, character13, character02, character16] **Bengali**: [character25, character18, character36, character16, character30, character15, character03, character42, character43, character31, character34, character08, character24] **ULOG**: [character09, character01, character11, character21] **Gurmukhi**: [character30, character14, character11, character44, character31, character27, character23, character36, character15, character40, character01] **Ojibwe_(Canadian_Aboriginal_Syllabics)**: [character07, character14] **Sanskrit**: [character34, character06, character30, character41, character12, character04, character05, character32] **Armenian**: [character24, character34, character41, character07, character31, character12, character28, character29, character20, character38, character23, character02, character03, character10, character06, character04] **Kannada**: [character06, character22, character03, character31] **Manipuri**: [character35, character37, character17, character39, character25, character28, character19, character01, character07, character40, character21, character08] **Anglo-Saxon_Futhorc**: [character22, character19, character28, character03, character23, character07, character15, character26, character12, character04] **Asomtavruli_(Georgian)**: [character36, character27, character37, character31, character01, character40, character19, character38, character13] **Keble**: [character25, character06, character02] **Japanese_(hiragana)**: [character03, character35, character15, character08, character43, character20, character18, character21, character29, character39, character48] **Latin**: [character12, character01, character11, character20, character17] **Hebrew**: [character08, character17, character04] **Avesta**: [character14, character12, character02, character04] **Glagolitic**: [character40, character24, character45, character20, character32, character43, character05, character02, character15, character27, character22, character34, character16] **Tengwar**: [character25, character01, character08, character20, character24, character23, character04, character17, character02, character07, character12] **Greek**: [character20, character15, character17, character16, character23, character11] **Sylheti**: [character02, character10, character07, character19, character15, character21, character11, character09] **Braille**: [character09, character16, character07, character08, character19, character06, character03] **Burmese_(Myanmar)**: [character01, character16, character11, character28, character17, character04, character02] **Futurama**: [character21, character23, character09, character18, character07] **Mongolian**: [character02, character09, character06, character05, character21, character04, character10] **Inuktitut_(Canadian_Aboriginal_Syllabics)**: [character01, character03, character04, character07] **N_Ko**: [character27, character07, character04, character22, character17, character08, character05, character29] **Blackfoot_(Canadian_Aboriginal_Syllabics)**: [character06, character02, character04, character14] **Balinese**: [character08, character10, character22, character16, character21] **Aurek-Besh**: [character04, character01] **Tifinagh**: [character12, character53, character51, character46, character55, character25, character22, character14, character47, character01, character27, character29] **Grantha**: [character10, character34, character29, character04, character05, character06, character43, character36, character28, character09, character41]

