# OpenReview forum: "MahiNet: A Neural Network for Many-Class Few-Shot Learning with Class Hierarchy"
_ICLR.cc/2019/Conference_

### Official Review · AnonReviewer2 · 2018-11-01
**An interesting paper which may need further enhancement**

**Rating:** 5
**Confidence:** 3

**Review:**

This study explores the class hierarchy to solve many-class few-short learning problem in both traditional supervised learning and meta-learning. The model integrates both the coarse-class and fine-class label as the supervision information to train the DNN, which aims to leverage coarse-class label to assist fine-class prediction. The core part in the DNN is memory-augmented attention model that includes at KNN classifier and Memory Update mechanism. The re-writable memory slots in KNN classifier aim to maintain multiple prototypes used to describe the data sub-distribution within a class, which is insured by designing the memory utility rate, cache and clustering component in Memory Update mechanism. This study presents a relatively complex system that combines the idea of matching networks and prototypical networks.

One of the contributions is that the study puts forward a concept of the many-class few-short learning problem in both supervised learning and meta-learning scenarios, and uses a dataset to describe this problem.

Using the memory-augmented mechanism to maintain multiple prototypes is a good idea. It may be more interesting if its effectiveness can be proved or justified theoretically. Furthermore, it is better to offer some discussion about the learned memory slots in the view of “diverse and representative feature”.

The experiment results in Table 4 and Table 5 compare the MahiNet with Prototypical Net on the mcfsImageNet and mcfsOmniglot dataset. It is better to compare MahiNet with other state-of-the-art works, such as the Relation Network whose performance is higher than Prototypical Net. In addition, if more  challenging datasets  can be further evaluated in the experiments,  the paper  might be more convincing.

In my opinion, the hierarchy information provides the guidance to fine-gained classification, which not only can be added to MahiNet but also the other models. Therefore, to prove its effectiveness, it is better to add hierarchy information to other models for comparison. In addition, regarding the results on the column of 50-5 and 50-10 in Table 4, when the number of class increase to 50, the results are just slightly higher than prototypical network. Considering that the memory update mechanism is of the high resource consumption and complexity, it is better to provide more details about clustering, and training and testing time.

---

> ### Author Response · Authors · 2018-11-26
> **New baselines with and without class hierarchy; t-SNE visualization reflects memory’s representativeness; Memory update is cheap (2)**
>
> Q4: The hierarchy information provides the guidance to fine-grained classification, which not only can be added to MahiNet but also the other models. Therefore, to prove its effectiveness, it is better to add hierarchy information to other models for comparison.
>
> R4: In the new experiments shown in Table 6 in Appendix D, we add the class hierarchy to relation network. As shown in the table, the class hierarchy improves the accuracy by more than 1%. We briefly list the experiment results as follows:
>
> The improvement of class hierarchy on relation network on mcfs-imagenet:
> ---------------------------------------------------------------------------------------------------
> Model                            Hierarchy             5 way 5 shot             5 way 10 shot
> Relation network               N                       63.02±0.87                 74.12±0.78
> Relation network               Y                        66.82±0.86                 75.31±0.90
> ---------------------------------------------------------------------------------------------------
> MahiNet                              Y                        74.98±0.75                 80.74±0.66
> ---------------------------------------------------------------------------------------------------
>
> Q5: Regarding the results on the column of 50-5 and 50-10 in Table 4, when the number of class increases to 50, the results are just slightly higher than prototypical network.
>
> R5: When the number of classes increases, the complexity of the task dramatically increases, and improving the performance of few-shot learning becomes much harder. Hence, comparing to >2-3% improvements on 20-way, the improvements on 50-way (>1% for 5-shot and >0.7% for 10-shot) are still significant considering the number of classes increases from 20 to 50. Note the reported performance of 50-way can be further improved by specifically tuning hyperparameters for 50-way, because the current hyperparameters are the same for 50-way, 20-way, and 5-way and achieved by tuning on 20-way experiments. We did not do a heavy tuning specifically for each setting due to our limited computational resources.
>
> Q6: Considering that the memory update mechanism is of the high resource consumption and complexity, it is better to provide more details about clustering, and training and testing time.
>
> R6: The time costs of memory update and clustering are negligible comparing to the total time costs, because we only keep and update a very small memory. For each epoch of supervised learning, the average clustering time is 30s and is only 7.6% of the total epoch time (393s). Within an epoch, the memory update time (0.02s) is only 9% of the total iteration time (0.22s). In meta-learning, we use a fixed memory storing all training samples and do not update it. So it has the same time cost as prototypical network [2].
>
> [1] Flood Sung Yang, Li Yongxin, Tao Xiang, Zhang, Philip HS Torr, and Timothy M. Hospedales. Learning to compare: Relation network for few-shot learning. In IEEE Conference on Computer Vision and Pattern Recognition (CVPR), 2018.
> [2] Jake Snell, Kevin Swersky, and Richard Zemel. Prototypical networks for few-shot learning. In Advances in Neural Information Processing Systems (NIPS), 2017.

---

> ### Author Response · Authors · 2018-11-26
> **New baselines with and without class hierarchy; t-SNE visualization reflects memory’s representativeness; Memory update is cheap (1)**
>
> Thanks for your comment! The following lists your questions and the corresponding replies:
>
> Q1: It is better to offer some discussion about the learned memory slots in the view of “diverse and representative feature”.
>
> R1: This is a good idea. In Figure 4 of Appendix-D, we visualize the feature vectors for the memory slots and the feature vectors of the images using T-SNE, where different colors represent different fine classes. It shows that for each class, the small number of feature vectors in memory are diverse and sufficiently representative of the whole class.
>
> Q2: It may be more interesting if its effectiveness can be proved or justified theoretically.
>
> R2: It is interesting but also challenging to provide a thorough theoretical analysis of the proposed memory augmented neural nets, considering that the theoretical properties of some basic neural networks without memory update are unclear. Intuitively, there is a trade-off: more prototypes lead to more representativeness, but too many will weaken generalization and efficiency. We will keep studying its theoretical properties in the future.
>
> Q3: It is better to compare MahiNet with other state-of-the-art works, such as Relation Network whose performance is higher than Prototypical Net. In addition, if more challenging datasets can be further evaluated in the experiments, the paper might be more convincing.
>
> R3: Thanks for the suggestion! We add relation network [1] as a new baseline in Table 4. Our method outperforms it in several different settings in the table. We briefly list the experiment results as follows:
>
> Comparison w.r.t. the accuracy (%) of different approaches in meta-learning scenario on mcfsImageNet. “n-k” represents n-way (class) k-shot.
> ----------------------------------------------------------------------------------------------------------
> Model                               Hierarchy                5-10                20-5                 20-10
> Prototypical network           N                 78.48±0.66      67.78±0.37      70.11±0.38
> Relation net                           N                 74.12±0.78      52.66±0.43     55.45±0.46
> ----------------------------------------------------------------------------------------------------------
> MahiNet                                 Y                  80.74±0.66      70.11±0.41     73.50±0.36
> ----------------------------------------------------------------------------------------------------------
>
> In our experiments, relation network fails when the number of classes (ways) is large: it usually stops at a suboptimal solution after the first several iterations (under different learning rates). We show this phenomenon in Figure 3 of Appendix-D by plotting the training loss and accuracy for the first 100 iterations. The loss and accuracy stay almost the same for iterations afterwards.
>
> One primary reason to build new datasets in this paper specifically for many-class few-shot learning problem is that the existing datasets are not challenging enough or do not fulfill the requirement of this problem. As shown by the comparison in Table 2, mcfsImageNet is the most challenging one for this problem since it has much more classes than others and the total number of images is large.

---

### Official Review · AnonReviewer1 · 2018-11-03
**Need to clarify some procedures**

**Rating:** 6
**Confidence:** 3

**Review:**

This paper try to formulate many-class-few-shot classification problem from 2 perspectives: supervised learning and meta-learning. Although solving this problem with class hierarchy is trivial,  combing MLP and KNN in these two ways seems interesting to me. I still have several questions:

1) How the class hierarchy is got , manually set or automatically generate?  Whether the ideas still work if some coarse classes share same fine class?
2) The so-called attention module is just classic KNN operations, please don't naming it attention just because the concept "attention" is hot.
3) Why different "attention" operations are used for supervised learning and meta-learning?
4) How to get the pre-trained models for supervised learning?
5) What will happen if alternatively apply supervised learning and meta-learning?
6) In Table 4, why MahiNet(Mem-2) w/o Attention and Hierarchy performs better than the one w/o attention?
7) The authors just compare storing the average features and all features, I think results of different prototype number should be given, since one of their claim to apply KNN is to maintain a small memory.

---

> ### Author Response · Authors · 2018-11-26
> **Exploring class hierarchy in few-shot learning is non-trivial; Clarity improved; More detailed explanations (2)**
>
> Q3: Why different "attention" operations are used for supervised learning and meta-learning?
>
> R3: We tried both options in Eq.(8) for the two learning scenarios in our experiments. According to our experience, while dot-product attention works better for supervised learning, Euclidean distance based attention is preferred in meta-learning (this is consistent with the observations in prototypical networks [2]).
>
> Q4: How to get the pre-trained models for supervised learning?
>
> R4: We train the CNN (ResNet)+MLP model by using the standard backpropagation to minimize the sum of cross entropy loss on coarse-classes and fine-classes.
>
> Q5: What will happen if alternatively apply supervised learning and meta-learning?
>
> R5: Theoretically, supervised learning and meta-learning have different optimization objectives, so alternatively applying the two might even increase the objective value of each (assuming each solves a minimization). In addition, our model is slightly different in these two settings (meta-learning uses a fixed memory and does not have memory update module), so alternatively applying the two is not even optimizing the same model structure.
>
> Intuitively, if the training/test sets of the tasks in meta-learning are sampled from the training/test set of supervised learning (which does not always hold in practice), training the model in supervised learning mode is helpful to improve the performance of meta-learning. However, this is a cheating and is not legal, because supervised learning exposes all the classes, which contain the classes of test tasks for meta-learning, but these classes should be unseen during training for meta-learning.
>
> Q6: In Table 4, why MahiNet(Mem-2) w/o Attention and Hierarchy performs better than the one w/o attention?
>
> R6: Since Table 4 does not include “MahiNet (Mem-2) w/o attention”, we assume that the reviewer refers to “MahiNet (Mem-2) w/o Hierarchy”. The main reason is that the many-class few-shot data without class hierarchy is not sufficient to learn a trustworthy attention (similarity metric), and such attention without sufficient training might be harmful to the final performance. In contrast, with class hierarchy, attention only needs to distinguish much fewer fine classes within each coarse class, and the learned attention (even on few-shot data) can faithfully reflect the local similarity within each coarse class.
>
> Q7: The authors just compare storing the average features and all features, I think results of different prototype number should be given, since one of their claims to apply KNN is to maintain a small memory.
>
> R7: In our experiments of supervised learning, the memory size is only 754*12(the number of memory features)/125321(the number of image features) = 7.2% of the dataset size. We also tried to increase the memory size to about 10%, but the improvement on performance is neglectable. In each task of meta-learning, since every class only has few-shot samples, the memory required to store all the data is not large. For example, in the 20-way-1-shot setting, the memory size is only 10.2KB, storing 20 features.
>
>
> [1] Mengye Ren, Eleni Triantafillou, Sachin Ravi, Jake Snell, Kevin Swersky, Joshua B. Tenenbaum, Hugo Larochelle and Richard S. Zemel. Meta-Learning for Semi-Supervised Few-Shot Classification. In International Conference on Learning Representations (ICLR), 2018.
> [2] Jake Snell, Kevin Swersky, and Richard Zemel. Prototypical networks for few-shot learning. In Advances in Neural Information Processing Systems (NIPS), 2017.

---

> ### Author Response · Authors · 2018-11-26
> **Exploring class hierarchy in few-shot learning is non-trivial; Clarity improved; More detailed explanations (1)**
>
> Thanks for your comments and suggestions! First of all, we want to emphasize that solving many-class few-shot problem with the class hierarchy is not trivial. It is not only about the design of the loss defined on the hierarchy, but also the model structure, i.e., which model should be used for coarse classification and which for fine classification, how to combine them in a unified model and how to let them cooperate with each other during training and inference. These are challenging problems with many possible options, but not every option leads to good performance.  We tried the published code with class hierarchy of [1] in both supervised few-shot learning and semi-supervised few-shot learning scenarios. However, all experiments failed to bring improvement in performance. In this paper, we carefully design new loss functions and network structures, which aim to maximize the assistance of the class hierarchy, and use the extra information provided by it to precisely solve the many-class few-shot problem.
>
> We are very pleased to answer other questions/comments from you one by one as follows.
>
> Q1: How the class hierarchy is got, manually set or automatically generate? Whether the ideas still work if some coarse classes share the same fine class?
>
> R1: In this paper, we use the class hierarchy provided by the original ImageNet and Omniglot datasets (details can be found in the 2nd and 3rd paragraphs of Appendix-A). In practice, the class hierarchy (the coarse class labels in specific) we required is usually available or cheap to achieve.
>
> It is interesting to study the case when some coarse classes share the same fine classes. The idea of this paper can be applied to this case. In particular, we only need to modify the right-hand side of Eq.(6) by summing over all possible coarse classes $z_i$, which is a result of applying Eq.(5) to the multiple coarse label case. This leads to a modified loss function. Most of the other parts of MahiNet can be kept the same.
>
> Q2: The so-called attention module is just classic KNN operations, please don't name it attention just because the concept "attention" is hot.
>
> R2: Attention module in MahiNet (Eq.(8)) is not classic KNN operations: it provides a learnable distance/similarity metric ($g()$ and $h()$ are learnable) for the KNN classifier (while classic KNN uses a fixed metric); and the produced similarities of each sample to its K nearest neighbors are used as weights to compute the final prediction of the sample. It plays exactly the same role as the attention modules used in other meta-learning models such as matching networks. Since learnable similarity and similarity weighted averaging are the two critical features of attention mechanism, it is more accurate to call it attention instead of KNN here.

---

### Official Review · AnonReviewer3 · 2018-11-12
**Class hierarchy and cache-based nearest neighbors for many class / few shot**

**Rating:** 5
**Confidence:** 3

**Review:**

This paper presents methods for (1) adding inductive bias to a classifier through coarse-to-fine prediction along a class hierarchy and (2) learning a memory-based KNN classifier through an intuitive procedure that keeps track of mislabeled instances during learning. Further, the paper motivates focused work on the many class / few shot classification scenario and creates new benchmark datasets from subsets of imagenet and omniglot that match this scenario. Experimental results show gains over popular competing methods on these benchmarks.
Overall, I like the motivation that this paper provides for many class / few shot and find some of the methods proposed interesting. Yet there are issues with clarity of presentation that made it somewhat difficult to fully understand the exact procedures that were implemented. The model figure is useful, but could be refined to add additional clarity -- particularly in the case of the KNN learning procedure.
I'm not entirely familiar with recent work in this sub-field, so it is difficult for me to judge the novelty of the proposed procedure. Is it really true that class-hierarchies have never been used to perform coarse-to-fine inference in past work? If so, this should be state clearly. If not, related work should be mentioned and compared against. Finally, while the procedures are intuitive -- the takeaway of this paper could be substantially improved if even simple theoretical analysis were provided. For example, in the limit of infinite data, does the memory-based KNN learning procedure actually produce the right classifier?
Misc comments questions:
-The paper says at least twice that coarse classification will be performed with an MLP, while fine classification will use a KNN -- yet, the model section also state that both coarse and fine use both MLP and KNN. It is unclear to me which model setup was used in experiments.
-Can anything theoretical be shown about the class hierarchy based classification technique? Intuitively, it does add inductive bias through a manually defined taxonomy, but can something more precise be said about how it restricts the hypothesis space? This procedure is simple enough that I would be surprised if similar techniques had not be studied thoroughly in the statistical learning theory literature.
-The procedure for updating the KNN memory is intuitive, but can anything more be said about it? In isolation, is the KNN learning procedure at least consistent -- i.e. in the limit of large data does it converge to the correct classifier? Maybe this is trivial to prove, but is worth including.

---

> ### Author Response · Authors · 2018-11-26
> **First success of applying class hierarchy to few-shot learning; More clarifications added; New discussions of several theoretical properties (2)**
>
> Q4: Can anything theoretical be shown about the class hierarchy based classification technique?...can something more precise be said about how it restricts the hypothesis space?
>
> R4: The high-level idea of this paper follows Neural Turing Machine [1], which is a well-known memory-based system. We follow NTM to create our memory-augmented mechanism, and make some specific modification to adapt the many-class few-shot problem.
>
> The hypothesis space is significantly reduced after applying class hierarchy. This can be shown by comparing Eq.(5) and Eq.(6), where Eq.(6) is derived from Eq.(5) after applying the class hierarchy. In particular, without class hierarchy, Eq.(5) will assign each fine class a nonzero probability, i.e., the hypothesis space is $C*F$ ($C$ is the number of coarse classes and $F$ is the number of fine classes in each coarse class), while Eq.(6) only assign nonzero probabilities to the fine classes within the coarse classes and rule out all the other fine classes, i.e., the hypothesis space is $F$.
>
> Q5: Simple theoretical analysis should be provided. For example, in the limit of infinite data, does the memory-based KNN learning procedure actually produce the right classifier?...The procedure for updating the KNN memory is intuitive, but can anything more be said about it? In isolation, is the KNN learning procedure at least consistent -- i.e. in the limit of large data does it converge to the correct classifier?
>
> R5: The memory updating procedure presented in Section 2.4 is a modified version of the coreset algorithm for streaming k-center clustering (Section 3 of [3]), where the distance metric for the clustering is the hamming distance between prediction and ground truth (0 if correct and 1 if wrong). In particular, if the prediction is correct, the new sample will be assigned to a feature vector in memory (i.e., a cluster centroid) and update it (Eq.(11)), otherwise the new sample will be written to the cache (Eq.(12)) as a new feature vector (to form new clusters). At the end of each epoch, according to the utility scores, we replace $r$ feature vectors in memory with the $r$ new cluster centroids computed from the cache, so the size of memory (the number of cluster centroids) keeps the same.
>
> If we assume that the correct classifier is the optimal (smallest error rate) KNN classifier with support set of limited size k (an NP-hard problem as k-center clustering), the theoretical properties of coreset guarantee that the support set achieved by our memory update procedure is a factor 8 approximation to the optimal KNN classifier on the hamming distance. However, since our training simultaneously updates the similarity matric (attention) and the input features to the KNN, the theoretical analysis becomes too complicated and the above result is not rigorous. So we are not sure if it is a good idea to include it in the paper.
>
>
> [1] Alex Graves, Greg Wayne, Ivo Danihelka. Neural Turing Machines. arXiv preprint arXiv:1410.5401, 2014.
> [2] Mengye Ren, Eleni Triantafillou, Sachin Ravi, Jake Snell, Kevin Swersky, Joshua B. Tenenbaum, Hugo Larochelle and Richard S. Zemel. Meta-Learning for Semi-Supervised Few-Shot Classification. In International Conference on Learning Representations (ICLR), 2018.
> [3] Moses Charikar, Chandra Chekuri, Tomás Feder, and Rajeev Motwani. 1997. Incremental clustering and dynamic information retrieval. In Proceedings of the twenty-ninth annual ACM symposium on Theory of computing (STOC '97).

---

> ### Author Response · Authors · 2018-11-26
> **First success of applying class hierarchy to few-shot learning; More clarifications added; New discussions of several theoretical properties (1)**
>
> Thanks for your comments and suggestions! You can find our response to each of your questions/comments in the following.
>
> Q1: The model figure is useful but could be refined to add additional clarity -- particularly in the case of the KNN learning procedure.
>
> R1: We have refined the figure to make it more clear. The learning of KNN classifier aims to optimize 1) the similarity metric parameterized by the attention (detailed in Section 2.3); and 2) a small support set of feature vectors per class stored in memory (detailed in Section 2.4). For a given sample, we compute the similarity scores of its feature vector to all the feature vectors from the support set per class by Eq. (8), and aggregate the scores to obtain a similarity score per class by Eq. (9). Then, a softmax is applied to compute probability over all classes from the similarity scores on all classes. During training, the attention block is updated by standard backpropagation, while the memory update procedure is detailed in line 5-9 and line 11-15 of Algorithm 1.
>
> Q2: Is it really true that class-hierarchies have never been used to perform coarse-to-fine inference in past work?
>
> R2: In this paper, class hierarchy is specifically used to make many-class few-shot learning possible. To the best of our knowledge, we are the first to successfully employ the class hierarchy information to improve few-shot learning. We tried the published code with class hierarchy of [2] in both supervised few-shot learning and semi-supervised few-shot learning scenarios. However, all experiments failed to bring improvement in performance. We have updated the manuscript to clarify this point.
>
> Q3: It is unclear to me which model setup was used in experiments.
>
> R3: As explicitly shown in Figure 2 and explained in the first paragraph of Section 2.2, in supervised learning we use MLP for coarse classification and MLP+KNN for fine classification, while in meta-learning we use MLP+KNN for coarse classification and MLP for fine classification. Intuitively, MLP performs better when data is sufficient (supervised learning), while the KNN classifier is more stable in few-shot scenario (meta-learning). Hence, we always apply MLP to coarse prediction, and apply KNN to fine prediction. In addition, we use KNN to assist MLP for coarse classification in meta-learning (when data might be insufficient even for coarse classes), and use MLP to assist KNN for fine classification in supervised learning (when data per fine-class is much more than the meta-learning case).

---

### Author Response · Authors · 2018-11-26
**Summary of Changes in Updated Draft**

We appreciate all the reviewers for their detailed comments and constructive suggestions. We also noticed all reviewers agreed that the problem and ideas presented in this paper are novel and interesting. We carefully updated the manuscript based on all the reviewers’ comments and highlighted all the modifications in the manuscript. Here is a summarization of our modifications :

1.	In order to improve clarity, we added more explanations and rewrote some parts of the paper where the reviewers asked for clarification. For example, we revised Figure 2, added the statement that we are the first to successfully apply the class hierarchy in few-shot learning, compared the time costs of memory update and clustering with the total time costs, and added more details about pre-training and analysis to experimental results.

2.	We did more experiments on a new baseline method. In particular, we compared to the original relation networks and relation networks with class hierarchy (in the same way as MahiNet). All the new results are reported in Table 4 and Appendix-D.

3.	We provide more explanation about memory update and KNN learning. In particular, we visualize the feature vectors selected into memory by T-SNE, and compare them with the rest feature vectors they are trying to represent. The 2D plot in Figure 4 of Appendix-E shows their diversity and representativeness. We also provide an analysis of memory usage at the end of the second paragraph in Section 4.1 and Appendix-E.

---

### Meta-Review · Area_Chair1 · 2018-12-20

**Confidence:** 4
**Recommendation:** Reject

**Metareview:**

The reviewers raised a number of concerns including low readability/ clarity of the presented work and methodology, insufficient and at times unconvincing experimental evaluation of the proposed, and lack of discussion on pros and cons of the presented. The authors’ rebuttal addressed some of the reviewers’ comments but failed to address all concerns and reconfirmed that relatively large changes are still needed for the paper to be useful to the readers. Hence, although I believe this could be a very interesting paper, I cannot suggest it at this stage for presentation at ICLR.